# SYSTEMATIC EVALUATION OF CAUSAL DISCOVERY IN VISUAL MODEL BASED REINFORCEMENT LEARNING

## ABSTRACT

Inducing causal relationships from observations is a classic problem in machine learning. Most work in causality starts from the premise that the causal variables themselves are observed. However, for AI agents such as robots trying to make sense of their environment, the only observables are low-level variables like pixels in images. To generalize well, an agent must induce high-level variables, particularly those which are causal or are affected by causal variables. A central goal for AI and causality is thus the joint discovery of abstract representations and causal structure. However, we note that existing environments for studying causal induction are poorly suited for this objective because they have complicated task-specific causal graphs which are impossible to manipulate parametrically (e.g., number of nodes, sparsity, causal chain length, etc.). In this work, our goal is to facilitate research in learning representations of high-level variables as well as causal structures among them. In order to systematically probe the ability of methods to identify these variables and structures, we design a suite of benchmarking RL environments. We evaluate various representation learning algorithms from the literature and find that explicitly incorporating structure and modularity in models can help causal induction in model-based reinforcement learning.

## 1 INTRODUCTION

Deep learning methods have made immense progress on many reinforcement learning (RL) tasks in recent years. However, the performance of these methods still pales in comparison to human abilities in many cases. Contemporary deep reinforcement learning models have a ways to go to achieve robust generalization (Nichol et al., 2018), efficient planning over flexible timescales (Silver & Ciosek, 2012), and long-term credit assignment (Osband et al., 2019). Model-based methods in RL (MBRL) can potentially mitigate this issue (Schrittwieser et al., 2019). These methods observe sequences of state-action pairs, and from these observations are able to learn a self-supervised model of the environment. With a well-trained world model, these algorithms can then simulate the environment and look ahead to future events to establish better value estimates, without requiring expensive interactions with the environment (Sutton, 1991). Model-based methods can thus be far more sample-efficient than their model-free counterparts when multiple objectives are to be achieved in the same environment. However, for model-based approaches to be successful, the learned models must capture relevant mechanisms that guide the world, i.e., they must discover the right causal variables and structure. Indeed, models sensitive to causality have been shown to be robust and easily transferable (Bengio et al., 2019; Ke et al., 2019). As a result, there has been a recent surge of interest in learning causal models for deep reinforcement learning (de Haan et al., 2019; Dasgupta et al., 2019; Nair et al., 2019; Goyal et al., 2019; Rezende et al., 2020). Yet, many challenges remain, and a systematic framework to modulate environment causality structure and evaluate models' capacity to capture it is currently lacking, which motivates this paper.

What limits the use of causal modeling approaches in many AI tasks and realistic RL settings is that most of the current causal learning literature presumes abstract domain representations in which the cause and effect variables are explicit and given (Pearl, 2009). Methods are needed to automate the inference and identification of such causal variables (i.e. *causal induction*) from low-level state

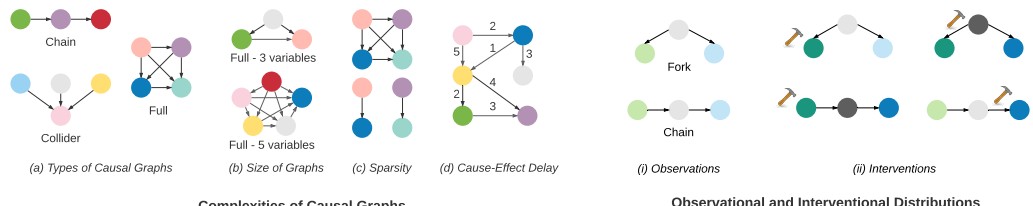

Figure 1: (a)-(d): Different aspects contributing to the complexity of causal graphs. (i), (ii): Difference between observational and interventional data. In RL setting, actions are interventions in the environment. The hammer denotes an intervention. Intervention on a variable not only affects its direct children, but also all reachable variables. Variables impacted by the intervention have a darker shade.

representations (like images). Although one solution is manual labeling, it is often impractical and in some cases impossible to manually label all the causal variables. In some domains, the causal structure may not be known. Further, critical causal variables may change from one task to another, or from one environment to another. And in unknown environments, one ideally aims for an RL agent that could induce the causal structure of the environment from observations and interventions.

In this work, we seek to evaluate various model-based approaches parameterized to exploit structure of environments purposfully designed to modulate causal relations. We find that modular network architectures appear particularly well suited for causal learning. Our conjecture is that causality can provide a useful source of inductive bias to improve the learning of world models.

***Shortcomings of current RL development environments, and a path forward.*** Most existing RL environments are not a good fit for investigating causal induction in MBRL, as they have a single fixed causal graph, lack proper evaluation and have entangled aspects of causal learning. For instance, many tasks have complicated causal structures as well as unobserved confounders. These issues make it difficult to measure progress for causal learning. As we look towards the next great challenges for RL and AI, there is a need to better understand the implications of varying different aspects of the underlying causal graph for various learning procedures.

Hence, to systematically study various aspects of causal induction (i.e., learning the right causal graph from pixel data), we propose a new suite of environments as a platform for investigating inductive biases, causal representations, and learning algorithms. The goal is to disentangle distinct aspects of causal learning by allowing the user to choose and modulate various properties of the ground truth causal graph, such as the structure and size of the graph, the sparsity of the graph and whether variables are observed or not (see Figure 1 (a)-(d)). We also provide evaluation criteria for measuring causal induction in MBRL that we argue help measure progress and facilitate further research in these directions. We believe that the availability of standard experiments and a platform that can easily be extended to test different aspects of causal modeling will play a significant role in speeding up progress in MBRL.

***Insights and causally sufficient inductive biases.*** Using our platform, we investigate the impact of explicit structure and modularity for causal induction in MBRL. We evaluated two typical of monolithic models (autoencoders and variational autoencoders) and two typical models with explicit structure: graph neural networks (GNNs) and modular models (shown in Figure 5). Graph neural networks (GNNs) have a factorized representation of variables and can model undirected relationships between variables. Modular models also have a factorized representation of variables, along with directed edges between variables which can model directed relationship such as $A$ causing $B$, but not the other way around. We investigated the performance of such structured approaches on learning from causal graphs with varying complexity, such as the size of the graph, the sparsity of the graph and the length of cause-effect chains (Figure 1 (a) - (d)).

The proposed environment gives novel insights in a number of settings. Especially, we found that even our naive implementation of modular networks can scale significantly better compared to other models (including graph neural networks). This suggests that explicit structure and modularity such as factorized representations and directed edges between variables help with causal induction in MBRL. We also found that graph neural networks, such as the ones from Kipf et al. (2019) are good at modeling pairwise interactions and significantly outperform monolithic models under this setting. However, they have difficulty modeling complex causal graphs with long cause-effect chains, such as the chain graph (demonstration of chain graphs are found in Figure 1 (i)). Another finding is that evaluation metrics such as likelihood and ranking loss do not always correspond to the performance of these models in downstream RL tasks.

Figure 2: Illustration of the key features of the suite. Environments have objects that interact according to the underlying causal graph which can be based on a subset of objects' properties. An efficient model should be able to infer the high level causal variables from raw pixel data and learn the underlying causal graph through interactions between these high level causal variables.

## 2 ENVIRONMENTS FOR CAUSAL INDUCTION IN MODEL-BASED RL

Causal models are frequently described using graphs in which the edges represent causal relationships. In these *structural causal models*, the existence of a directed edge from $A$ to $B$ indicates that intervening on $A$ directly impacts $B$, and the absence of an edge indicates no direct interventional impact (see Appendix A for formal definitions). In parallel, world models in MBRL describe the underlying data generating process of the environment by modeling the next state given the current state-action pair, where the actions are interventions in the environment. Hence, learning world models in MBRL can be seen as a causal induction problem. Below, we first outline how a collection of simple causal structures can capture real-world MBRL cases, and we propose a set of elemental environments to express them for training. Second, we describe precise ways to evaluate models in these environments.

### 2.1 MINI-ENVIRONMENTS: EXPLICIT CASES FOR CAUSAL MODULATION IN RL

The ease with which an agent learns a task greatly depends on the structure of the environment's underlying causal graph. For example, it might be easier to learn causal relationships in a collider graph ( see Figure 1(a)) where all interactions are pairwise, meaning that an intervention on one variable $X_i$ impacts no more than one other variable $X_j$, hence the cause-effect chain has a length of at most 1. However, causal graphs such as full graphs (see Figure 1 (a)) can have more complex causal interactions, where intervening on one variable impacts can impact up to $n-1$ variables for graphs of size $n$ (see Figure 1). Therefore, one important aspect of understanding a model's performance on causal induction in MBRL is to analyze how well the model performs on causal graphs of varying complexity.

Impotant factors that contribute to the complexity of discovering the causal graph are the *structure*, *size*, *sparsity of edges* and *length of cause-effect* chains of the causal graph (Figure 1). Presence of *unobserved variables* also adds to the complexity. The size of the graph increases complexity because the number of possible graphs grows super-exponentially with the *size of the graph* (Eaton & Murphy, 2007; Peters et al., 2016; Ke et al., 2019). The *sparsity of graphs* also impacts the difficulty of learning, as observed in (Ke et al., 2019). Given graphs of the same size, denser graphs are often more challenging to learn. Futhermore, the *length of the cause-effect* chains can also impact learning. We have observed in our experiments, that graphs with shorter cause-effect lengths such as colliders (Figure 1 (a)) can be easier to model as compared to chain graphs with longer cause-effect chains. Finally, *unobserved variables* which commonly exist in the real-world can greatly impact learning, especially if they are confounding causes (shared causes of observed variables).

Taking these factors into account, we designed two suites of (toy) environments: the *physics environment* and the *chemistry environment*, which we discuss in more detail in the following section. They are designed with a focus on the underlying causal graph and thus have a minimalist design that is easy to visualize.

#### 2.1.1 PHYSICS ENVIRONMENT: WEIGHTED-BLOCK PUSHING

The physics environment simulates very simple physics in the world. It consists of blocks of different, unique weights. The rule for interaction between blocks is that heavier objects can push lighter ones. Interventions ammount to move a particular block, and the consequence depends on whether the block next to it (if present) is heavier or lighter. For an accurate world model, inferring the weights becomes essential. Additionally, one can allow the weight of the objects to be either observed through

Figure 3: Demonstration of the weighted-block pushing environment (left: observed, right: unobserved) along with the feasible generalizations that the setup provides.

the intensity of the color, or unobserved, leading to two environment settings described below. The underlying causal graph is an acyclic tournament, shown in Figure 3. For more details about the setup, please refer to Appendix E.

*Fully observed setting.* In the fully observed setting, all objects are given a particular color and the weight of each block is represented by the intensity of the color. Once the agent learns this underlying causal structure, it does not have to perform interventions on new objects in order to infer they will interact with the others.

*Unobserved setting.* In this setting, the weight of each object is not directly observable by its color. The agent thus needs to interact with the object in order to understand the order of weights associated with the blocks. In this case, the weight of objects needs to be inferred through interventions. We consider two sub-divisions of this setting - *FixedUnobserved* where there is a fixed assignment between the shapes of the objects and their weights and *Unobserved* where there is no fixed assignment between the shape and the weight, hence making it a more challenging environment. We refer the reader to Appendix E.2 for details.

### 2.1.2 CHEMISTRY ENVIRONMENT

The chemistry environment enables more complexity in the causal structure of the world by allowing arbitrary causal graphs. This is depicted by simple chemical reactions, where the state of an element can cause changes to another variable's state. The environment consists of a number of objects whose positions are kept fixed and thus, uniquely identifiable.

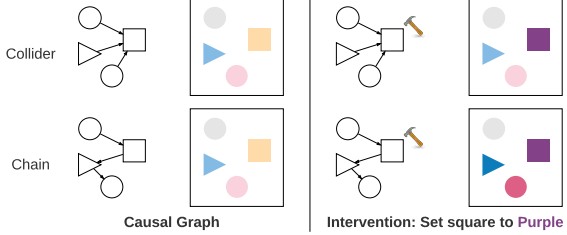

Figure 4: Demonstration of the vanilla chemistry environment (left: ground truth causal graph and a sample from it - same sample shown to demonstrate the affect of interventions, right: the affect of interventions and how far they affect based on underlying causal graph)

The interactions between different objects take place according to the underlying causal graph which can either be a randomly generated DAG, or specified by the user. An interaction consists of changing the color (state) of a variable. At this point, the color of all variables affected by this variable (according to the causal graph) can change. Interventions change a block's color unconditionally, thus cutting the graph edge linking it with its parents in the graph. All transitions are probabilistic and defined by conditional probability tables (CPTs). A visualization of the environment can be found in Figure 4.

This environment allows for a complete and thorough testing of causal models as there are various degrees of complexities which can be easily tuned such as: (1) Complexity of the graph: We can test any model on many different graphs thus ensuring that a models performance is not only limited to a few select graphs. (2) Stochasticity: By tuning the skewness of the probability distribution of each object we can test how good is a given model in modelling data uncertainty. In addition to this we can also tune the number of object or the number of colors to test whether the model generalizes to larger graphs and more colors. A causally correct model should be able to infer the causal relationships between observed objects, as well as their respective color distribution and its dependence on a causal parent's distribution.

## 2.2 EVALUATING CAUSAL MODELS

In much of the existing literature, evaluation of learned causal models is based on the structural difference between the learned graph and the ground-truth graph (Peters et al., 2016; Zheng et al., 2018). However, this may not be applicable for most deep RL algorithms, as they do not necessarily learn an explicit causal structure (Dasgupta et al., 2019; Ke et al., 2020). Even if a structure is learned, it may not be unique as several variable permutations can be equivalent, introducing an additional evaluation burden. Another possibility is to exhaustively evaluate models on all possible intervention predictions and all environment states, a process that quickly becomes intractable even for small environments. We therefore propose a few evaluation methods that can be used as a surrogate metrics to measure the model's performance on recovering the correct causal structure.

*Predicting Intervention Outcomes.* While it may not be feasible to predict all intervention outcomes in an RL environment, we propose that evaluating predictions on a subset of interventions provides an informative evaluation. Here, the test data is collected from the same environment used in training, ensuring a single underlying causal graph. Test data is generated from new episodes that are unseen during training. All interventions (actions) in the test episodes are randomly sampled and we evaluate the model's performance on this test set.

*Zero Shot Transfer.* Here, we test the model's ability to generalize to unseen test environments, where the environment does not have exactly the same causal graph as training, but training and test causal graphs share some similarity. For example, in the *observed* Physics environment, a model that has learned the underlying causal relationship between color intensity and weight would be able to generalize to new variables with a novel color intensity.

*Downstream RL Tasks.* Downstream RL tasks that require a good understanding of the underlying causal graph of the environment are also good metrics for measuring the model's performance. For example, in the *physics environment*, we can provide the model with a target configuration in the form of some specific arrangement of blocks on a grid and the model needs to perform actions in the environment to reach the target configuration. Models that capture causal relationships between objects should achieve the target configuration more easily (as it is can predict intervention outcomes). For more details about this setup, please refer to Appendix C.

*Metrics.* We also evaluate the learned models on ranking metrics in the latent space as well as reconstruction-based metrics in the observation space (Kipf et al., 2019). In particular we measure and report Hits at Rank 1 (H@1), Mean Reciprocal Rank (MRR) and Reconstruction loss for evaluation in standard as well as transfer testing settings. We report these metrics for 1, 5 and 10 steps of prediction in the latent space (refer Appendix B).

## 3 MODELS

A large variety of neural network models have been proposed as world models in MBRL. These models can roughly be divided into two categories: *monolithic models* and models that have *structure* and *modularity*. *Monolithic models* typically have no explicit structure (other than layers). Some typical monolithic models are Autoencoders and Variational Autoencoders (Kingma & Welling, 2013; Rezende et al., 2014). Conversely, *structured* models have explicit architecture built into (or learned by) the model. Examples of such models are ones based on graph neural networks (Battaglia et al., 2016; Van Steenkiste et al., 2018; Kipf et al., 2019; Veerapaneni et al., 2020) and modular models (Ke et al., 2020; Goyal et al., 2019; Mittal et al., 2020;

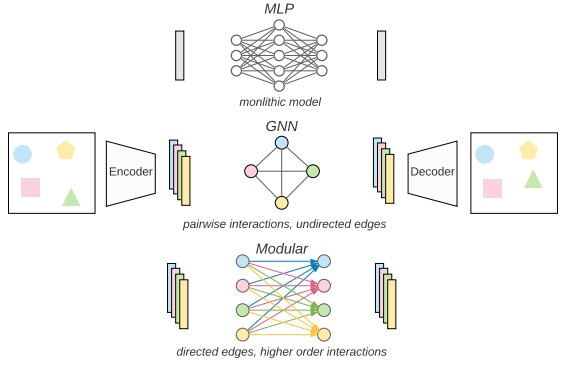

Figure 5: All models have 3 components: *encoder*, *decoder* and *transition model*. The transition models can either be monolithic, modular models or graph neural networks (GNNs). Monolithic models don't have explicit structure. GNNs have factorized representation of variables. Modular models have both factorized representation of variables and directed edges to potentially model causal relationships such as $A$ causing $B$.

Goyal et al., 2020). We picked some commonly used models from these categories and evaluated their performance to understand their ability for causal induction in MBRL.

To disentangle the architectural biases and effects of different training methodologies, we trained all the models on both likelihood based and contrastive losses, respectively. All models share three common components: *encoder*, *decoder* and *transition model*. We follow a similar training procedure as in Ha & Schmidhuber (2018); Kipf et al. (2019). Details of the architectures as well as the training protocols and losses can be found in Appendix D.

### 3.1 MONOLITHIC MODELS

We evaluate causal induction on two commonly used monolithic models: multilayered autoencoders and variational autoencoders. We follow a similar setup as in Ha & Schmidhuber (2018). These models do not have strong inductive biases other than the number of layers used.

### 3.2 MODULAR AND STRUCTURED MODELS

Several forms of structure can be included in neural networks, including *modularity*, *factorized variables*, and *directed rules*. Taking the three factors into account, we consider two types of structured models in our paper, *graph neural networks* (GNN) and so called *modular networks*. Graph neural networks (GNN) (Gilmer et al., 2017; Tacchetti et al., 2018; Battaglia et al., 2018; Kipf et al., 2019) is a widely adopted relational model that have a factorized representation of variables and models pairwise interactions between objects while being permutation invariant. In particular, we consider the C-SWM model (Kipf et al., 2019), which is a state-of-art GNN used for modeling object interactions. Similar to most GNNs, the C-SWM model learns factorized representations of different objects but for modelling dynamics it considers all possible pairwise interactions, and hence the transition model is monolithic (i.e., not a modular transition model).

Modular networks on the other hand are composed of an initial encoder that factorizes inputs (images), and then a *modular transition model* (MTM) - $M$. This internal model is tasked to create separate factored representations for each objects in the environment, while taking into account all other objects' representations. This model also learns interactions between objects. The rules learned here are *directed rules*.

## 4 EXPERIMENTS

Our experiments seak to answer the following questions: (a) Does explicit structure and modularity help for causal induction in MBRL? If so, then what type of structures provide good inductive bias for causal induction in MBRL? (b) How do different objective functions (likelihood or contrastive) impact learning? (c) How do different models scale to complex causal graphs? (d) Do prediction metrics (likelihood and ranking metrics) correspond to better downstream RL performance? (e) What are good evaluation criteria for causal induction in MBRL?

We report the performance of our models on both the Physics and the Chemistry environments, and refer the readers to Appendix D for implementation details.. All models are trained using the procedure described in Section D.2 and are evaluated based on *ranking* and *likelihood metrics* on 1, 5 and 10 step predictions. For the Chemistry environment, we evaluate the models on causal graphs with varying complexity, namely - *chain*, *collider* and *full* graphs. These graphs vary in *the sparsity of edges* and the *length of cause-effect chains*. For the Physics environment, we evaluate the model in the fully observed setting as well as the unobserved setting.

### 4.1 EXPLICIT STRUCTURE AND CAUSAL INDUCTION

We found that for both the Physics and the Chemistry environments, models with explicit structure outperform monolithic models on both prediction metrics and downstream RL performances. In particular, models with explicit structure (GNNs and modular models) scale better to graphs of *larger size* and *longer cause-effect chains*.

The Physics environment has a complex underlying causal graph (full graph: refer Figure 1 (a)). We found that GNNs performed well in this environment with 3 variables. They achieved good prediction metrics (Figure 7) and high RL performance (Figure 13) even at longer timescales. However, their performance drops significantly on environments with 5 objects both in terms of prediction metrics (Figure 8) and RL performance (Figure 14). We also see in Figure 8 and 14 that modular models scale much better compared to all other models, suggesting that they hold an advantage for *larger*

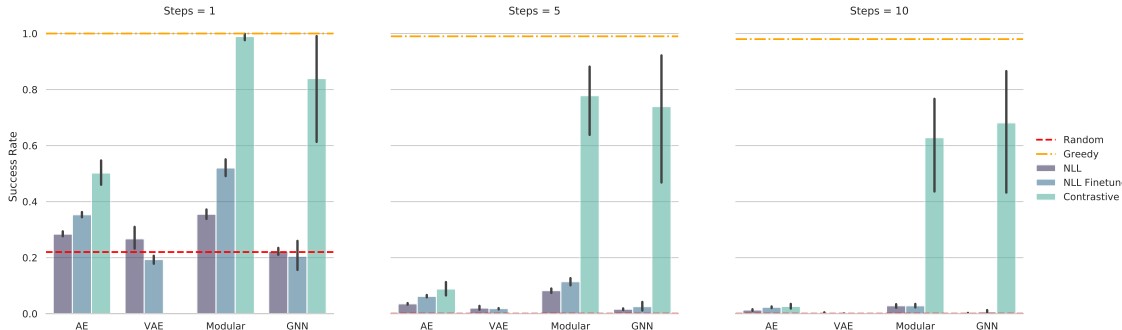

Figure 6: Success Rate *(higher is better)* for different models and training losses for 1, 5 and 10 step prediction for the Fixed Unobserved Physics environment setting with 5 objects. Here, (a) Random stands for a random policy, (b) greedy is the policy with best greedy actions, (c) NLL are models trained in 2 stages: pretraining the encoder/ decoder, following by only training the transition model, (d) NLL with finetune are models in 3 stages: pretraining the encoder/ decoder, following by only training the transition model and then finetuning the encoder, decoder and transition model together. (e) Contrastive are models trained using a contrastive loss. The GNN and Modular models trained on constrastive loss significantly outperform the monolithic models (autoencoders and VAE). The margin significantly increases as the number of steps to reach the goal increase, suggesting that models with explicit structure and modularity have a much better understanding of the world.

causal graphs. Further, modular models and GNNs when evaluated on zero shot settings outperform monolithic models by a significant margin (Figures 19, 20 and Tables 15, 16).

For the chemistry environment, we find that modular models outperform all other models for almost all causal graphs in terms of both prediction metrics (Fig 23) and RL performance (Fig 25). This is especially true on more complex causal graphs, such as *chain* and *full* graphs which have long cause-effect chains. This suggests that modular models scales better to more complex causal graphs.

Overall, these results suggest that structure, and in particular modularity, help causal induction in MBRL when scaling up to larger and more complex causal graphs. The performance comparisons on modular networks and C-SWM (Kipf et al., 2019) suggest that both factorized representation of variables and directed edges between variables can help for causal induction in MBRL.

### 4.2 Complexity of the Underlying Causal Graph

There are several ways to vary complexity in a causal graph: *size of the graph*, *sparsity of edges* and *length of cause-effect chain* (Figure 1). Increasing the size of the graph significantly impacts all models' performances. We evaluate models on the Physics environments with 3 objects (Figure 7) and 5 objects (Figure 8) and find that increasing the number of objects from 3 to 5 has a significant impact on performance. Modular models achieve over 90 on ranking metrics over 10-step prediction for 3 objects while for 5 objects, they achieve only 50 (almost half the performance on 3 objects). A similar pattern is found in almost all models. Another factor impacting complexity of the graph is the *length of cause-effect chain*. We see that collider graphs are the easiest to learn, with modular models and autoencoders significantly outpeforming all other models (Figure 23). This is because the collider graph has short pair-wise interactions, i.e, intervention on any node in a collider graph can impact at most one other node. Chain and full graphs are significantly more challenging because of longer cause-effect chains. For a chain or a full graph of $n$ nodes, an intervention on the $k^{th}$ node can impact all the subsequent $(n - k)$ nodes. Modeling interventions on chain and full graphs require modeling more than pairwise relationships, hence, making it much more challenging. We find that modular models slightly outperform all other models on these graphs.

### 4.3 Prediction Metrics and RL Performance

As discussed in Section 2.2, there are multiple evaluation metrics based on either prediction metrics or RL performance. The performance of the model on one metric may not necessarily transfer to another. We would like to analyze if this is the case for the models trained under various environments. We first note that while the ranking metrics were relatively good for most models on physics environments, most of them only did slightly better than a random policy on downstream RL, especially on larger graphs (Figures 7 - 12 and Tables 3 - 8 for ranking metrics; Figures 13 - 18 and Tables 9 - 14 for downstream RL). Figures 21, 22 and 27 show scatter plots for each pair of losses, with one loss on each axis. While there is some correlation between ranking metric and RL performance (Modular

and GNN; Figure 21), we did not find this trend to be consistent across models and environment settings. We feel that these results give further evidence of need to evaluate on RL performance.

## 4.4 Training objectives and learning

Likelihood loss and contrastive loss (Oord et al., 2018; Kipf et al., 2019) are two frequently used objectives for training world models in MBRL. We trained the models under each of these objective functions to understand how they impact learning. In almost all cases, models with explicit structure (modular models and GNNs) trained on contrastive loss perform better in terms of ranking loss compared to those trained on likelihood loss (refer to Figures 7 - 12). We don't see a very clear trend between training objective and downstream RL performance but we do see a few cases where contrastively trained models performed much better than others (refer to Figures 6, 13, 17, 18 and Tables 9, 13, 14).

For other key insights and experimental conclusions on different environments, we refer the readers to Appendix E.6 for the physics environment and Appendix F.3 for the chemistry environment.

## 5 Related work

*Video Prediction and Visual Question Answering.* There exist a number of video prediction (Yi et al., 2019; Baradel et al., 2019) and visual question answering (Johnson et al., 2017) datasets that also make use of a blocks world for visual representation. Though these datasets can appear visually similar to ours at first glance, they lack two essential ingredients for systematically evaluating models for causal induction in MBRL. The first is that they do not allow active interventions and hence make it challenging for evaluating model-based reinforcement learning algorithms. Another key point is that these environments do not allow one to systematically perturb different aspects of causal graphs, hence, preventing to systematically study the performances of models for causal induction.

*RL Environments.* There exist several benchmarks for multi-task learning for robotics (Meta-World (Yu et al., 2019) and RLBench (James et al., 2020)) and for video gaming domain (Arcade Learning Environment, CoinRun (Cobbe et al., 2018), Sonic Benchmark (Machado et al., 2018), MazeBase (Nichol et al., 2018) and BabyAI (Chevalier-Boisvert et al., 2018)). However, as mentioned earlier, these benchmarks do not allow one to systematically controll different aspects of causal models (such as the structure, the sparsity of edges and the size of the graph), hence making it difficult to systematically study causal induction in MBRL.

*Block World.* The AI community has been using the "blocks world" for decades as a testbed for various AI problems, including learning theory (Winston, 1970), natural language (Winograd, 1972), and planning (Fahlman, 1974). Block world allows to easily vary different aspects of the underlying causal structure, and also allow interventions to be performed on many high level variables of the environment giving rise to a large space of tasks which have well-defined relations between them.

## 6 Discussions and conclusions

In our work, we focus on studying various model-based approaches for causal induction in model-based RL. We highlighted the limitations of existing benchmarks and introduced a novel suite of environments that can help measure progress and facilitate research in this direction. We evaluated various models under many different settings and discuss the essential problems and challenges in combining both fields i.e ingredients, that we believe are common in the real world, such as modular factorization of the objects and interactions of objects governed by some unknown rules. Using a proposed evaluation framework, we demonstrate that structural inductive biases are beneficial to learning causal relationships and yield significantly improved performances in learning world models.

There are several interesting future directions that can be taken from here. One direction is extending the environments to settings such as meta-learning, where different causal graphs are set for each episode of training. Another interesting direction is extending this to an environment where the cause and effect does not happen at fixed timescale. For example, if a person smokes, it can take variable amount of time until they get cancer. This is very relevant for reinforcement learning, as this is tightly related to credit assignment in RL.

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
