# OpenReview forum: "Systematic Evaluation of Causal Discovery in Visual Model Based Reinforcement Learning"
_ICLR.cc/2021/Conference — Reject_

### Official Review · AnonReviewer4 · 2020-10-28
**Another benchmark to facilitate causal induction for RL with pixel-level observations.**

**Rating:** 6
**Confidence:** 3

**Review:**

Summary:

This paper proposes a suite of RL benchmarks to facilitate inducing causal relationships from visual observations. They show that structural inductive biases are beneficial for causal relationship learning and model-based RL by testing a variety of representation learning algorithms on this benchmark.

##########################################################################

pros:

+ The motivation is clear and interesting. Inducing causal relationships from pixel-level observations is an important topic to integrate causality into RL.

+ Overall, the paper is well written.

+ A comprehensive evaluation is conducted to highlight the usage of the proposed benchmark.

##########################################################################

cons:

- Would you explain more about the physics environment about why the underlying graph is a causal graph, not just a statistical graphical model?

- Why is PHYRE[1] benchmark not mentioned in the related work? You seem to work in the same direction and they have proposed more complicated and interesting tasks.

[1] Bakhtin, A., van der Maaten, L., Johnson, J., Gustafson, L., & Girshick, R. (2019). Phyre: A new benchmark for physical reasoning. In Advances in Neural Information Processing Systems (pp. 5082-5093).


##########################################################################

Post rebuttal

I'm happy with the author's response and would like to keep my original score.

---

> ### Author Response · Authors · 2020-11-15
> **Thank you for your review.**
>
> We thank the reviewer for their review, and we are enthused that the reviewer find the paper "well-written" and "interesting".
>
> > Would you explain more about the physics environment about why the underlying graph is a causal graph, not just a statistical graphical model?
>
> The physics environment is  designed to model simple causal relations in real-world such as simple physics. It is defined by “directed” rules that heavy blocks can push lighter blocks, but not vice versa. Hence, the actions (interventions) of heavier objects can “cause” the positions of the lighter objects, but not the other way around, and this makes the relationships between objects of different weights “causal”. The underlying causal graph is an acyclic tournament, ordered from the heaviest object to the lightest.
>
> > “Why is PHYRE[1] benchmark not mentioned in the related work? You seem to work in the same direction and they have proposed more complicated and interesting tasks.”
>
> Thanks for pointing out this reference, we will certainly discuss and cite the paper in the updated version of our paper. We agree that there are similarities between PHYRE and our work. The main difference is PHYRE does not allow experimenters to set arbitrary causal graphs for the environment, whereas  our environments allow for much more flexibility in defining the underlying causal graph. We feel that the flexibility of setting the underlying causal graph is important for systematically studying causal induction in MBRL, hence, making our contributions valuable.

---

### Official Review · AnonReviewer3 · 2020-10-29
**Software for reviewing performance of integrating causality to RL in a number of application-driven environments.**

**Rating:** 4
**Confidence:** 3

**Review:**

This paper is a review of model-based approaches of integrating causal inference to reinforcement learning (RL) in different environments (application areas). The authors provide software to analyse how three types of models (“monolithic”, i.e. latent space models without a graph-like structure of the latent space, graph neural networks (GNN) and “modular”, i.e. the C-SWM model (Kipf et al., 2020)) perform in two artificial “environments” devised by the authors (physics and chemistry) based on a number of metrics, some of them also proposed by the authors. The main contributions are the platform for evaluating models in the environments and the insights from the experiments performed on the selected models (taken from existing literature).


*****Strengths:*****

The paper is very well-written, clear and easy to follow.

The paper provides a novel perspective on applications of causality modelling and the presented environments could be useful to practitioners in the respective fields of physics and chemistry.


*****Weaknesses:*****

As the paper has the form of a review and a collection of insights, there seems to be no novelty from the point of view of either machine learning, causal inference or reinforcement learning. I am therefore not sure whether ICLR is the right venue for this paper.

While the motivation for the causal problem tackled in the paper (inferring model-driven causal insights usable for RL agents from data available to agents/robots such as images) is well-defined, interesting and relevant, there seems to be little actual follow-up on it in the paper. The presented models do not seem to adhere to any causal formalism (Pearl’s graphs, Rubin’s potential outcomes) and their only “causal” interpretation is a naïve use of structures (graphs and DAGs) as descriptors of causal relationships without any formal justification. There appears to be no relation to causal structure learning. Neither GNN nor C-SWM were conceived as causal models, and if one wishes to use them as causal models, more justification for such use is needed (why should the discovered graphs correspond to some underlying causal relationships?). Moreover, there have been approaches to achieve what this paper seems to promise (discover causal relationships from images), e.g. (Discovering causal signals in images, Lopez-Paz et al., 2017).


*****Post Rebuttal*****
I would like to thank the authors for the detailed rebuttal.

The authors state: "The main goal of our paper is NOT to introduce novel models, but rather to introduce a NOVEL benchmark and insights/ingredients to study causal induction in model-based RL" and "It is true that the models we use do not learn an explicit structure for causal learning" which corresponds to my original reservations to the novelty of this paper. The authors introduce a benchmark / software for evaluating causal induction in RL models, where the user can specify a causal model and its influence on the environment can be examined. I remain unconvinced that the introduction of a RL evaluation benchmark (even one allowing for the presence of arbitrary causal networks) counts as novel at ICLR.

Further, the statement "we used some of the typically common models for this purpose, such as GNNs and modular networks. Though these models do not learn an explicit causal graph, they do learn structure that could allow them to discover causal relationships under certain assumptions" confirms my point that no causal formalism (e.g. connection to the data generating process) is accounted for. It merely means that directed relationships can be modelled.

I agree with the authors that (Lopez-Paz et al., 2017) only uses observational data and does not have any connection to RL, but it is an example of an approach to extracting causal relationships from images in a sound way when it comes to causal inference. This is a side comment and does not influence my assessment of the paper.

To sum up, my reservations towards the degree of novelty in this paper have not changed (I agree with the authors' summary of their contributions, but disagree as to whether proposing a new benchmark constitutes novelty at ICLR) and I recommend a rejection of the paper in its current form.

---

> ### Author Response · Authors · 2020-11-15
> **Causal Structure Learning is a hard problem!**
>
> We thank the reviewer for their feedback. We are happy that the reviewer find the proposed work well-written and easy to follow.
>
> > “There seems to be no novelty from the point of view of either machine learning, causal inference or reinforcement learning. I am therefore not sure whether ICLR is the right venue for this paper.“
>
> The contribution of our paper lies in the intersection of causal induction and reinforcement learning. Classic causal induction methods assume that the causal variables are meaningful and given to the algorithm and this is generally not the case in model-based RL. More so, existing RL environments does not allow systematic evaluation of causal induction in MBRL as they 1) have a fixed causal graph, 2) do not allow for arbitrary interventions on causal variables.
>
> The main goal of our paper is NOT to introduce novel models, but rather to introduce a NOVEL benchmark and insights/ingredients to study causal induction in model-based RL, which to the best of our knowledge does not exist in current literature.  Hence, we hope our work  can help to facilitate research for causal induction in MBRL.
>
> Our major contributions can be summarized as follows
>
>
> The main purpose of our paper is NOT to introduce novel architecture (models),  but rather to introduce a benchmark for causal learning in model-based RL, which to our knowledge does not exist in current literature. On top of that, we show that current methods fail to achieve good performance on this benchmark.  Our major contributions can be summarized as follows:
>
> - Most existing work in causality starts from the premise that the causal variables themselves are observed. This is seldom the case in real-world settings, where the observables are mostly low-level variables such as pixels. In such scenarios it is important for the model to first correctly extract the underlying causal variables from the pixel space. Keeping this in mind we design a suite of environments such that raw pixels are fed into the models and it should learn to extract the correct variables and their underlying relations from these pixels.
>
> - In order to systematically evaluate the ability of different methods to  learn high level representations, and causal relations between these variables, we propose a suite of environments as existing benchmarks are not sufficient for this problem. In existing benchmarks (MetaWorld, Phyre, Atrari, CLEVRER etc.) the amount of shared causal structure between the different environments is mostly unknown, e.g. in the Atari Arcade Learning environments, it is unclear how to quantify the underlying similarities between different Atari games and we generally do not know to which degree an agent can be expected to generalize. The graphs in these environments are exceedingly complicated for even humans to discern concretely. For example, we may have a high-level understanding of the causal structure in any ATARI game but it is extremely difficult to concretely define this structure using a graph and other related parameters. If we want to evaluate the causal learning ability of a model, it is first important that we completely understand the causal relations in the environment!
>
> - Most current environments provide NO WAY to manipulate the underlying causal graph in their environments. Therefore, the graphs in environments like ATARI and CLEVRER are FIXED. As opposed to this, the causal graphs in our environments are NOT FIXED. We provide the user complete control over the underlying causal graph in the environment such that the user can specify a plethora of different graphs by varying parameters like number of nodes, number of edges etc. This is extremely important as it gives us the capacity to test any model on a wide range of different graphs before claiming that it succeeds in capturing causal relations. Hence, we designed a set of benchmarks and RL environments  that allow the experimenter to systematically perturb different aspects of the RL environment and hence better understand and evaluate agents’ generalization ability with respect to different types and extents of changes in the environment.
>
> - We investigate the impact of explicit structure and modularity for causal induction in MBRL. We evaluated two typical monolithic models (autoencoders and variational autoencoders) and two typical models with explicit structure: graph neural networks (GNNs) and modular models (shown in Figure 5). Graph neural networks (GNNs) have a factorized representation of variables and can model undirected relationships between variables. Modular models also have a factorized representation of variables, along with directed edges between variables which can model directed relationships such as A causing B, but not the other way around.
> To the best of our knowledge,  we are not aware of any other work which tries to evaluate different models in a systematic way for causal structure learning in the context of model based RL.

---

> > ### Author Response · Authors · 2020-11-15
> > **GNN is NOT Causal model, limitations of Lopez-Paz et al., 2017 for Model-based RL**
> >
> > > “There appears to be no relation to causal structure learning. Neither GNN nor C-SWM were conceived as causal models, and if one wishes to use them as causal models, more justification for such use is needed (why should the discovered graphs correspond to some underlying causal relationships?).”
> >
> > It is true that the models we use do not learn an explicit structure for causal learning (to our knowledge no such model exists till now).  However, learning structures for a causal model is not the only way to evaluate a model for causal induction, we can also evaluate the model’ ability for causal induction by  evaluating it to predict interventional outcomes (as in our task).
> >
> > Since there are no existing ready to use methods for causal-induction in MBRL, we used some of the typically common models for this purpose, such as GNNs and modular networks. Though these models do not learn an explicit causal graph, they do learn structure that could allow them to discover causal relationships under certain assumptions.  Graph neural networks (GNNs) have a factorized representation of variables and can model undirected relationships between variables. Modular models also have a factorized representation of variables, along with directed edges between variables which can model directed relationships such as A causing B, but not the other way around.  A main contribution of our paper is to highlight that there are indeed LIMITATIONS for existing models on causal learning and hence stressing the importance for research in this area.
> >
> > > “Moreover, there have been approaches to achieve what this paper seems to promise (discover causal relationships from images), e.g. (Discovering causal signals in images, Lopez-Paz et al., 2017).”
> >
> > There are several aspects in the paper of Lopez-Paz et al., 2017 that limits its use in model-based RL.
> > - The work of Lopez-Paz et al., 2017 uses only observational data, not interventional data. However, in model-based RL, the objective of the world model is to predict interventional outcomes. Hence, this work is NOT suitable to be used for MBRL.
> >
> > - Moreover, there are theoretical limitations on the identifiability of underlying structures obtained from observational data alone. Interventional data provides much richer information about the underlying data-generating process. There exists methods designed for causal discovery from interventional data [1, 2, 3], however, they assume that the right causal variables are meaningful and given, which is not the case for RL. Hence, this makes it challenging to apply such methods for MBRL. If the reviewer has ideas on how to use Lopez-Paz et al., 2017 or other methods for MBRL, we would be happy to run those experiments.
> >
> > - [1] Learning Neural Causal Models from Unknown Interventions, https://arxiv.org/abs/1910.01075
> > - [2] Causal Discovery from changes, https://arxiv.org/abs/1301.2312
> > - [3] Causality in Machine Learning, https://arxiv.org/abs/1911.10500
> >
> > We hope this answers your questions. Please let us know if there's something we can do to increase your confidence as well as score. Thanks.

---

> > > ### Author Response · Authors · 2020-11-17
> > > **Anything else you'd like us to respond to?**
> > >
> > > Hello,
> > >
> > > We thank the reviewer for their feedback and valuable comments.  Since the first phase of response period is completed, if you have time and could indicate if there are any other concerns of yours which we have not addressed, we'd be happy to take a look.
> > >
> > > Thanks for your time.

---

### Official Review · AnonReviewer1 · 2020-10-29
**Right direction; writing is confusing; just comparing existing methods, but no new method**

**Rating:** 4
**Confidence:** 3

**Review:**

This paper investigates the importance of incorporating structure and modularity in MBRL. Specifically, it compares monolithic models with models with explicit structure: GNNs and modular models. It also investigates the influences of varying complexity of graphs.

Pros:
1. The authors state a general goal in the abstract: "the joint discovery of abstract representations and causal structure", which is promising and needs more investigation for sure.

Cons:
1. The writing is a bit confusing. From the title and the abstract, the readers may expect that this paper is about causal discovery, i.e., learning casual relationships from data, while the casual structure actually is given in the main text.
2. In addition, this paper only compares existing models with or without structures, but I fail to see any novel or interesting ideas that this paper tries to deliver.
3. Furthermore, from the experiments, the authors found that increasing the size of the graph impacts the performance significantly, e.g., increasing 3 nodes to 5 nodes. This result surprises me. I think if the structure is handled properly, the increase from 3 nodes to 5 nodes should not have such a big effect. The authors may investigate or propose other methods that make proper use of the structures.

Based on the above reasons, I do not think this paper is ready to publish.


Post-rebuttal:
Thanks for the feedback. The goal of learning causal representation is ambiguous, and it is absolutely a good research topic. However, I fail to see an obvious contribution of the current version. Researchers in this field are usually clearly aware of the limitations for existing methods in causal learning. The problem is how to handle it, e.g., how to give a appropriate definition of the causal variables, how to theoretically show the identifiability and consistency, and then propose a practical solution.

---

> ### Author Response · Authors · 2020-11-15
> **Causal structure is NOT known to the learner (Part 1/2)**
>
> We are grateful that the reviewer finds our work promising.
>
> > “learning causal relationships from data, while the causal structure actually is given in the main text.”
>
> The environment has an underlying true causal graph based on which the data is generated, but the models DO NOT have access to the underlying causal graph, the models only see samples from the causal graph. The model would have to learn to discover the underlying causal structure (graph).
>
> > “ I think if the structure is handled properly, the increase from 3 nodes to 5 nodes should not have such a big effect. The authors may investigate or propose other methods that make proper use of the structures.”
>
> We agree with the reviewer that this would be true if the causal structure is GIVEN to the model. However, in our task setup, the model has to LEARN to discover the causal structure and the difficulty of this problem scales exponentially with the size of the graph. A main contribution of our paper is indeed to highlight that existing techniques have difficulty LEARNING the right causal structure.

---

> > ### Author Response · Authors · 2020-11-17
> > **Anything else you'd like us to respond to?**
> >
> > Hello,
> >
> > We thank the reviewer for their feedback and valuable comments.
> >
> > Since the first phase of response period is completed, if you have time and could indicate if there are any other concerns of yours which we have not addressed, we'd be happy to take a look.
> >
> > Thanks for your time.

---

> ### Author Response · Authors · 2020-11-15
> **Contributions of the proposed work: Evaluating Causal Models is an OPEN problem (Part 2/2)**
>
> "In addition, this paper only compares existing models with or without structures, but I fail to see any novel or interesting ideas that this paper tries to deliver."
>
>
> Our contributions:
>
> The main purpose of our paper is NOT to introduce novel architecture (models),  but rather to introduce a novel benchmark for causal learning in model-based RL, which to our knowledge does not exist in current literature. On top of that, we show that current methods fail to achieve good performance on this benchmark.  Our major contributions can be summarized as follows
>
> -  Most existing work in causality starts from the premise that the causal variables themselves are observed. This is seldom the case in real-world settings, where the observables are mostly low-level variables such as pixels. In such scenarios it is important for the model to first correctly extract the underlying causal variables from the pixel space. Keeping this in mind we design a suite of environments such that raw pixels are fed into the models and it should learn to extract the correct variables and their underlying relations from these pixels.
> -  In order to systematically evaluate the ability of different methods to  learn high level representations, and causal relations between these variables, we use a suite of environments as existing benchmarks are not sufficient for this problem. In existing benchmarks (MetaWorld, Phyre, Atrari, CLEVRER etc.) the amount of shared causal structure between the different environments is mostly unknown, e.g. in the Atari Arcade Learning environments, it is unclear how to quantify the underlying similarities between different Atari games and we generally do not know to which degree an agent can be expected to generalize. The graphs in these environments are exceedingly complicated for even humans to discern concretely. For example, we may have a high-level understanding of the causal structure in any ATARI game but it is extremely difficult to concretely define this structure using a graph and other related parameters. If we want to evaluate the causal learning ability of a model, it is first important that we completely understand the causal relations in the environment!
> - Most current environments provide NO WAY to manipulate the underlying causal graph in their environments. Therefore, the graphs in environments like ATARI and CLEVRER are FIXED. As opposed to this, the causal graphs in our environments are NOT FIXED. We provide the user complete control over the underlying causal graph in the environment such that the user can specify a plethora of different graphs by varying parameters like number of nodes, number of edges etc. This is extremely important as it gives us the capacity to test any model on a wide range of different graphs before claiming that it succeeds in capturing causal relations. Hence, we designed a set of benchmarks and RL environments  that allow the experimenter to systematically perturb different aspects of the RL environment and hence better understand and evaluate agents’ generalization ability with respect to different types and extents of changes in the environment.
> - We investigate the impact of explicit structure and modularity for causal induction in MBRL. We evaluated two typical monolithic models (autoencoders and variational autoencoders) and two typical models with explicit structure: graph neural networks (GNNs) and modular models (shown in Figure 5). Graph neural networks (GNNs) have a factorized representation of variables and can model undirected relationships between variables. Modular models also have a factorized representation of variables, along with directed edges between variables which can model directed relationships such as A causing B, but not the other way around.
> To the best of our knowledge,  we are not aware of any other work which tries to evaluate different models in a systematic way for causal structure learning in the context of model based RL.

---

### Official Review · AnonReviewer2 · 2020-11-08
**This paper introduces an interesting benchmark with an ambitious goal, but the tasks may be a bit too contrived.**

**Rating:** 5
**Confidence:** 4

**Review:**

=== Summary

This paper proposes a benchmark that aims to systematically evaluate models' ability in learning representations of high-level variables as well as causal structures among them. The authors introduce two benchmarking RL environments:
- One is in a physical domain where an agent is pushing blocks of different weights.
- Another one is to simulate a chemistry environment, where the state of an element can cause changes to another variable's state according to the underlying causal graph.

The authors evaluate several representation learning algorithms from the literature and find that explicitly incorporating structure and modularity in models can help causal induction in model-based reinforcement learning.


=== Strengths

This paper targets an important problem of assessing models' ability to automate the inference and identification of the causal variables from high-dimensional inputs like images.

The construction of the benchmark allows building causal graphs with varying complexity, such as the size of the graph, the sparsity of the graph, and the length of cause-effect chains.

The authors have evaluated several baseline models on the benchmark, including two typical monolithic models (autoencoders and variational autoencoders) and two models with explicit structure: graph neural networks (GNNs) and modular networks.

They have made several interesting observations, e.g., modular networks hold better scalability than other baselines, suggesting the benefits that explicit structure and modularity bring for causal induction in MBRL.


=== Weaknesses

My primary concern of this paper is that the dataset is a bit too contrived, which makes it hard to know whether the observations from this benchmark can generalize to more complicated real-world scenarios.

For example, in the Physics Environment proposed in this paper, only heavier objects can push lighter ones, not the other way around. I understand the authors' desire to make the underlying graph a directed acyclic graph (DAG), but it does not reflect what will happen in the real world. One could imagine sliding a lighter object to collide with a heavier one; the motions of both objects are likely to change, where the interaction between them is bi-directional.

Also, in the Chemistry Environment, a few objects are connected by a randomly generated DAG, where interventions can change the color of the target and the subsequent blocks. In chemistry, a molecule is a group of atoms held together by chemical bonds, which are also bi-directional relationships. Are there any specific examples in chemistry where the graph is a DAG? I would appreciate it if the authors can elaborate on the connection between the design of the environment and "chemistry."

In terms of the difficulty of the tasks, the results shown in Figure 6 suggest that the Greedy algorithm can achieve a near-perfect performance on the tasks, which consistently outperforms all other baselines. If a simple greedy algorithm can solve the tasks, does it mean that the benchmark may be a bit too simple, where a good understanding of the underlying causal structure may not be necessary? It would be better if the authors can discuss the necessity of causal induction in these tasks, and how is the ability to perform causal inference correlate with the metrics used in the benchmark.

Overall, I feel the environments proposed in the paper are a bit too artificial, which does not reflect what's likely to happen in the real world. While I like the goal of this paper, I think a set of more realistic environments could greatly improve the significance and potential impact of this paper.



=== Other comments


The font size of the image caption may be a bit too small.

Typo: Section 2.1, "Impotant" --> "Important"


=== Post rebuttal

The authors' rebuttal addressed some of my concerns, but my primary concern still remains that that benchmark may be a bit too contrived, where the observations made in this paper may not generalize to more complicated real-world situations. The authors also made some far-fetched arguments in the rebuttal by claiming some concurrent works [1, 2] as "the 'real-world' version of the environments used in the paper," which, to be honest, further lowers the rating of the paper on my side: why is this paper worthy of acceptance if there exist more realistic benchmarks?

I also agree with R1 and R3 that there are no new methods proposed in the paper, and the insights derived from benchmarking a set of existing methods may not be considered novel from the point of view of the ICLR audience. As a result, I keep my rating the same.

[1] Physically Embedded Planning Problems: New Challenges for Reinforcement Learning, https://arxiv.org/abs/2009.05524

[2] CausalWorld: A Robotic Manipulation Benchmark for Causal Structure and Transfer Learning, https://arxiv.org/abs/2010.04296

---

> ### Author Response · Authors · 2020-11-15
> **Tasks are simple, but CURRENT Methods FAIL to learn the underlying causal structure even on SIMPLE tasks (Part 1/2)**
>
> > “My primary concern of this paper is that the dataset is a bit too contrived.”
>
> There's a trade off between being "real worldly" and allowing you to systematically manipulate the environment. Our main goal is to facilitate research in causal learning in RL environments, where the model has to handle 2 aspects of the problem, 1) learn to map from high-dimensional pixels to the abstract representations of causal variables, 2) learn the causal structures between these variables. It has been shown in model-based RL that learning both aspects is challenging, hence, the focus of our paper is to build a framework where we can focus on the second aspect of the problem (learning the causal structure of the environment).
>
> Even though the environment may seem visually simple, the underlying causal discovery problem is non-trivial. In fact,  this is a challenging environment (tasks) for many common methods (models) used in MBRL.
>
> Moreover, our suite of environments allow the experiment to systematically evaluate models (methods) for causal induction in MBRL. More specifically, our suite of environments have following properties:
>
> - Allow for perturbation on different aspects of causal graphs, such as the number of variables, number of edges etc, such that any model that claims to capture causal relations can be thoroughly evaluated on a plethora of different types of graphs.
> - The ability to intervene on different variables in the causal graph.
> - The environment should share some causal structure to allow algorithms to transfer  the learned causal structure from one environment to another.
> - Previous work has only evaluated the ability of learned representations by using some proxy of maximum likelihood or mutual information. The goal of the current work is to also evaluate the learned representations for downstream RL tasks.
>
> > “generalize to more complicated real-world scenarios”
>
> We note parallel to our work, [1], [2] introduced a set of physically embedded problems for evaluating the ability of the current deep learning algorithms to reason over long time horizons. They consider tasks similar to the ones introduced in our work like the Physics env, and then embed them into a physical setting where in order to execute a move, the agent is required to control a physical boxy over many simulation time-steps.  They note that even though the current RL algorithms can tackle the symbolic version of these tasks (sokoban, tictactoe, and go), they struggle to master even the simplest of their physically embedded counterparts. The environment of MuJoBan introduced in [1] is like taking the physics env in the current work, and then physically embedding it. (They run the method for 100M steps). This shows that even though the environments are simple, there’s already some evidence that the “real-world” version of the environments used in the paper would still be a challenge for current algorithms.
>
> - [1] Physically Embedded Planning Problems: New Challenges for Reinforcement Learning, https://arxiv.org/abs/2009.05524
> - [2] CausalWorld: A Robotic Manipulation Benchmark for Causal Structure and Transfer Learning, https://arxiv.org/abs/2010.04296
>
>
> Inducing causal relationships from observations is a classic problem in machine learning. Most work in causality starts from the premise that the causal variables themselves are observed. However, for AI agents such as robots trying to make sense of their environment, the only observables are low-level variables like pixels in images. To generalize well, an agent must induce high-level variables, particularly those which are causal or are affected by causal variables. A central goal for AI and causality is thus the joint discovery of abstract representations and causal structure. However, we note that existing environments for studying causal induction are poorly suited for this objective because they have:
>
> - Complicated task-specific causal graphs for which it is difficult for even humans to concretely discern the underlying causal variables and their relations. For example, for games like atari we may have a high level understanding of the causal structure but it is extremely difficult for us to clearly define that causal structure using a graph and some related parameters.
>  - It is impossible to manipulate the causal graphs in these environments  parametrically (e.g., number of nodes, sparsity, causal chain length, etc.).
>
> In this work, our goal is to facilitate research in learning representations of high-level variables as well as causal structures among them. In order to systematically probe the ability of methods to identify these variables and structures, we design a suite of benchmarking RL environments. We evaluate various representation learning algorithms from the literature and find that explicitly incorporating structure and modularity in models can help causal induction in model-based reinforcement learning.

---

> > ### Author Response · Authors · 2020-11-15
> > **Greedy Algorithm uses True Dynamics and True Reward Functions: (Part 2/2)**
> >
> > > “ I understand the authors' desire to make the underlying graph a directed acyclic graph (DAG), but it does not reflect what will happen in the real world. “
> >
> > We agree that this is a simplified version of what happens in the real-world. We kept our environment simple so that we can easily perturb different aspects of the underlying causal graph in the environment,  we also evaluated and demonstrated the challenges for causal induction and generalization even in such simple environments.
> >
> > > “In terms of the difficulty of the tasks, the results shown in Figure 6 suggest that the Greedy algorithm can achieve a near-perfect performance on the tasks, which consistently outperforms all other baselines”
> >
> > The greedy algorithm, which constructs a one step greedy policy, uses TRUE transitions and reward functions, they serve as an upper bound to how well our models could perform (if it had learned the true transition and reward model). In reality,  we do not have access to the true transition model or the true reward function. The aim of the greedy policy is to highlight how well an optimal causal model can perform on a fixed greedy policy IF it correctly learns the causal structure and transition dynamics.
> >
> > > “Also, in the Chemistry Environment, a few objects are connected by a randomly generated DAG, where interventions can change the color of the target and the subsequent blocks. In chemistry, a molecule is a group of atoms held together by chemical bonds, which are also bi-directional relationships. Are there any specific examples in chemistry where the graph is a DAG? I would appreciate it if the authors can elaborate on the connection between the design of the environment and "chemistry."”
> >
> > We agree with the reviewers that the reactions in a real chemistry environment are bidirectional. We apologize if the name might be misleading. We named the environment “chemistry” due to the similarity to how chemical elements change their state(molecular composition) after undergoing a reaction, the objects in our environment also change their state(color) after undergoing an intervention. We do not intend to model chemistry in terms of atoms or bonds between them. We use a DAG to specify the causal relations between the various objects or entities.
> >
> > In chemistry one common type of reaction is when an element comes in contact with a catalyst it changes its state. We can think of an intervention on the causal graph as  a catalyst and the object on which the intervention takes place changes its state(in this case color) akin to an element in chemistry. The “reactions” in our environment can be thought of as chain reactions as when one object changes its state, other objects depending on the causal graph also change their state.
> >
> > The main purpose of having the chemistry environment is to give users a lot of flexibility in terms defining the causal graph. This is akin to chemistry as in chemistry also an element can go through many number of possible reactions conditioned on the existence of right circumstances(for example: existence of the right catalyst, right elements etc. ). This flexibility is not possible in the physics environment as the physics environment comes with a fixed rule that only heavy blocks can push lighter blocks. This potentially limits the number of states that an object in the physics environment can take as well gives limited flexibility in defining the causal graph. In the chemistry environment, an object can take many possible states and there is no limitation on the causal relationships that we can define for the objects as long as they follow a DAG.

---

> > > ### Author Response · Authors · 2020-11-19
> > > **Anything else reviewer would like to see ?**
> > >
> > > Hello,
> > >
> > > We thank the reviewer for their feedback and valuable comments.
> > >
> > > Since the first phase of response period is over, if you have time and could indicate if there are any other concerns of yours which we have not addressed, we'd be happy to take a look.
> > >
> > > We are willing to spend time and efforts in order to improve the paper.
> > >
> > > Thanks for your help and time.

---

### Author Response · Authors · 2020-11-25
**Rebuttal Summary**

We thank all the reviewers for their time, and feedback. We believe we have addressed most of  the concerns of all the reviewers.
 We are glad that R2, R3, R4 found the paper to be very well written.

Here's a summary of our rebuttal:

__Problem__:  Inducing causal relations is a classic problem. Most work in causality starts from the premise that the causal variables themselves have known semantics or are observed. However, for AI agents such as robots trying to make sense of their environment, the only observables are low-level variables like pixels in images. To generalize well, an agent must induce high-level variables, particularly those which are causal or are affected by causal variables. A central goal for AI and causality is thus the joint discovery of abstract representations and causal structure. In this work, we systematically evaluate the agent's ability to learn underlying causal structure.

__Existing Benchmarks are not suitable__ : We note that existing environments (atari, meta-world etc.) for studying causal induction are poorly suited for this  objective because they have complicated task-specific causal graphs with many confounding factors.  It is impossible to manipulate the causal graphs in these environments parametrically (e.g., number of nodes, sparsity, causal chain length, etc.). Hence, to facilitate research in learning the representation of high-level variables as well as causal structure among these  variables, we present a suite of RL environments created to systematically probe the ability of methods to identify variables as well as causal structure among those variables.

__Essential Ingredients__:  Most of the worlds we care about have structure:  in most worlds, there are distinct entities, entities have possibly hidden or abstract properties that can determine their behavior, and entities interact with one another in a manner that depends on their properties. This description applies to video games, the physical world, social interactions, etc. Our primary conjecture is that a description of an environment that explicates the representations of entities and their properties---e.g., a factorial  representation---will allow for more robust feature learning.

__Modular models scale better than GNNs__: We evaluated two typical of monolithic models (autoencoders and variational autoencoders) and two typical models with explicit structure: graph neural networks (GNNs) and modular models (shown in Figure 5). Graph neural networks (GNNs) have a factorized representation of variables and can model undirected relationships between variables. Modular models also have a factorized representation of variables, along with directed edges between variables which can model directed relationship such as A causing B, but not the other way around. We show that the modular models *scale* better as compared to graph neural networks when we increase the complexity of the problem by increasing number of nodes, sparsity, causal chain length etc.).

__Dataset is a bit too contrived__:  A main contribution of our paper is to highlight that there are indeed limitations for existing models for causal learning even on such simple tasks.  Moreover [1, 2] introduced a set of physically embedded problems and consider tasks similar to the ones introduces in our work for evaluating the ability of the current deep learning algorithms to reason over long time horizons.  They note that the current methods struggle to master even the simplest of their physically embedded counterparts (i.e., in real world).

__Causal graph is NOT given to the learner__: The learner does not have access to the underlying causal graph.

__Greedy Baseline__:  The greedy algorithm, which constructs a one step greedy policy, uses TRUE transitions and reward functions, they serve as an upper bound to how well our models could perform.


[1] Physically Embedded Planning Problems: New Challenges for Reinforcement Learning, https://arxiv.org/abs/2009.05524
[2] CausalWorld: A Robotic Manipulation Benchmark for Causal Structure and Transfer Learning, https://arxiv.org/abs/2010.04296

---

### Decision · Program_Chairs · 2021-01-07
**Final Decision**

**Decision:**

Reject

**Comment:**

This paper proposes a suite of benchmark visual model-based RL tasks to evaluate causal discovery approaches under systematically varying causal graphs. Despite some disagreement on this point among reviewers, I would come down on the side of saying that a better-executed version of this paper would have been a good fit at ICLR. However, its current drawbacks make this a borderline reject. The most important of these drawbacks is: it is unclear to what extent results on these simple environments translate to more realistic complex ones.

Reviewers have also pointed to omitted relevant work that could be discussed in future versions, such as PHYRE. Another relevant benchmark in this vein: https://arxiv.org/abs/1907.09620